# Time-Averaged Hematuria as a Prognostic Indicator of Renal Outcome in Patients with IgA Nephropathy

**DOI:** 10.3390/jcm11226785

**Published:** 2022-11-16

**Authors:** Mengjie Weng, Jiaqun Lin, Yumei Chen, Xiaohong Zhang, Zhenhuan Zou, Yi Chen, Jiong Cui, Binbin Fu, Guifen Li, Caiming Chen, Jianxin Wan

**Affiliations:** 1Department of Nephrology, Blood Purification Research Center, The First Affiliated Hospital of Fujian Medical University, Chazhong Road 20, Fuzhou 350005, China; 2Fujian Clinical Research Center for Metabolic Chronic Kidney Disease, The First Affiliated Hospital of Fujian Medical University, Fuzhou 350005, China

**Keywords:** IgA nephropathy, persistent hematuria, predictors, risk factors, prognosis

## Abstract

We aim to investigate the association of time-averaged hematuria (TA-hematuria) with the progression of IgA nephropathy (IgAN). Based on TA-hematuria during follow-up, 152 patients with IgAN were divided into a hematuria remission group (≤28 red blood cells [RBCs]/μL) and a persistent hematuria group (>28 RBCs/μL). The persistent hematuria group had a higher percentage of patients with macroscopic hematuria, lower levels of hemoglobin and TA-serum albumin, and more severe renal pathologic lesions. The composite endpoint is defined as a doubling of the baseline SCr level (D-SCr), or the presence of ESRD. During the mean follow-up of 58.08 ± 23.51 months, 15 patients (9.9%) reached the primary outcome of ESRD and 19 patients (12.5%) reached the combined renal endpoint. Kaplan-Meier analysis showed that the persistent hematuria group had a lower renal survival rate. The persistent hematuria patients who were incorporated with proteinuria (≥1.0 g/day) and low TA-serum albumin (<40 g/L) had the worst renal outcomes. Multivariate Cox regression indicated that TA-hematuria (hazard ratio [HR] = 0.004, 95% CI: 0.001, 0.008; *p* = 0.010) was independently associated with the progression of IgAN. Receiver operating characteristic analysis indicated the optimal TA-hematuria cutoff value for predicting the progression of IgAN was 201.21 RBCs/μL in females and 37.25 RBCs/μL in males.

## 1. Introduction

IgA nephropathy (IgAN) is the most common primary glomerular disease in China and worldwide, and microscopic hematuria with proteinuria and episodic gross (macroscopic) hematuria are the most common clinical manifestations in these patients [1,2,3]. Although persistent microscopic hematuria is not a specific symptom of IgAN, IgAN accounts for half of all glomerular diseases detected by renal biopsies of patients who present with microscopic hematuria [4]. Other research concluded that IgAN greatly increased the risk of end-stage renal disease (ESRD) within 10 to 20 years from its apparent onset [2]. Consequently, there is an urgent need to identify predictors of outcome in patients with IgAN and to evaluate novel treatments that can slow disease progression. 

Large, multi-center studies have identified several independent risk factors for predicting the renal prognosis of patients with IgAN, and these include hypertension, urine protein level at the time of renal biopsy, estimated glomerular filtration rate (eGFR), and histopathological grade [3,5]. However, it is uncertain whether hematuria is a risk factor of renal outcome in IgAN. For many years, clinicians considered hematuria as an indicator of dysfunctional glomerular filtration, and a benign condition that was associated with IgAN [6]. Some cohort studies suggested that hematuria was unrelated to the long-term prognosis of patients with IgAN [7,8,9], but other studies concluded that hematuria was associated with IgAN progression [10,11,12]. These conflicting results may be due to the use of different measurement protocols. For example, some studies based their results on a single measurement of urine sediment at the time of the renal biopsy. 

Recent clinical studies suggested that persistent hematuria, determined as time-averaged hematuria (TA-hematuria) during follow-up, was a strong risk factor for reduced eGFR or kidney failure in patients with IgAN [3,13,14]. However, no detailed investigation has yet assessed the value of persistent hematuria as a predictor of poor prognosis in patients with IgAN, or the most appropriate timing for interventions in these patients. In this study, we analyzed the clinicopathological characteristics of patients who had IgAN and microscopic hematuria and determined the association of long-term kidney outcomes in patients who had persistent hematuria or remission from hematuria during follow-up. We also evaluated the value of TA-hematuria for predicting the progression of IgAN.

## 2. Materials and Methods

### 2.1. Participating Patients 

This cohort study initially examined patients who had microscopic hematuria and were diagnosed with IgAN based on renal biopsy results from June 2012 to December 2018 at the Department of Nephrology, First Affiliated Hospital of Fujian Medical University (Fuzhou, China). The exclusion criteria were the number of glomeruli in renal biopsy less than 8, secondary IgAN (caused by systemic lupus erythematosus, Henoch-Schonlein purpura, chronic liver disease, etc.), hemorrhagic illnesses, anticoagulant overdose, infectious illnesses, and endocrine metabolic illnesses (such as diabetic nephropathy, etc.), receipt of renal biopsy when less than 18 years old, and follow-up less than 12 months. All 152 participating patients were followed-up until June 2021 (Figure 1). 

### 2.2. Follow-Up and Data Collection

Each patient presented for regular visits at 6-month intervals, and proteinuria, urine sediment, serum albumin, and serum creatinine (SCr) were determined. The use of medications during follow-up, including renin-angiotensin-aldosterone system (RAAS) inhibitors, glucocorticoids, and immunosuppressants (including tacrolimus, cyclophosphamide, and cyclosporine), were recorded.

The following clinical data were collected at the time of renal biopsy: sex, age, serum albumin, SCr, eGFR, blood urea nitrogen (BUN), uric acid, total cholesterol (TC), triglycerides (TG), serum C3, serum C4, 24 h urinary protein, urinary red blood cells, history of gross macroscopic hematuria, and hypertension. After biopsy, all of the lesions were characterized using the Oxford classification (MEST-C) [15] as mesangial hypercellularity (M0/1), endocapillary hypercellularity (E0/1), segmental glomerulosclerosis (S0/1), tubular atrophy/interstitial fibrosis (lesions ranging from 0 to 25% as T0 and ˃25% as T1–T2), or crescent lesions (no crescents as C0, and presence of crescents as C1–C2).

### 2.3. Definitions

TA-hematuria was determined as the area under the curve from microscopic measurements of red blood cells (RBCs/μL) during follow-up divided by the months of total follow-up [3] (Appendix A). All of the measurements were performed using the same automated method in the same laboratory. The time-averaged serum albumin (TA-serum albumin) was determined using the same method. All patients had TA-hematuria at baseline. At the end of follow-up, patients were considered to have persistent hematuria if their urine had more than 28 RBCs/μL and they were otherwise considered to have hematuria remission (≤28 RBCs/μL) [14].

Hypertension was defined by a systolic blood pressure of 140 mmHg or more or diastolic blood pressure of 90 mmHg or more, based on measurements performed at rest on 3 different days, or by use of an antihypertensive medication to achieve control of blood pressure. Gross hematuria was defined by the visual presence of tea-colored, cola-colored, pink, or red urine. The diagnostic criteria for microscopic hematuria were 5 or more RBCs per high-power field (HPF) of a urine specimen, and females were instructed to avoid testing during menstruation. Patients were diagnosed with proteinuria remission if the urine protein level was 1.0 g/day or less and there was a decline of more than 25% from the baseline value [16]; otherwise, patients were diagnosed with persistent proteinuria. A low TA-serum albumin (or persistent hypoalbuminemia) was defined as a TA-serum albumin below 40 g/L. The eGFR was calculated using the Chronic Kidney Disease Epidemiology (CKD-EPI) equation [17]. 

The composite endpoint is defined as a doubling of the baseline SCr level (D-SCr), or the presence of ESRD (eGFR < 15 mL/min/1.73 m^2^).

### 2.4. Statistical Analysis

Data analysis was performed using SPSS-25 software (IBM Corp., Armonk, NY, USA). Categorical variables are presented as numbers and percentages, and the significance of differences was determined using the Chi-square test or Fisher’s exact test. Continuous variables are presented as means ± standard deviations (SDs) or as medians and interquartile ranges (IQRs), and the significance of differences was determined using a *t*-test or the Mann-Whitney *U* test. Different groups were compared using Kaplan-Meier survival curves, and these curves were compared using the log-rank test. Multivariate Cox regression analysis was used to determine the significance of the relationship of different risk factors with patient prognosis. Receiver-operating characteristic (ROC) analysis was used to determine the value of persistent hematuria in predicting the progression of IgAN. Statistical significance was defined by a two-sided *p*-value less than 0.05.

## 3. Results

### 3.1. Patient Characteristics

We retrospectively examined the records of 152 patients who had IgAN based on renal biopsy results (Table 1). There were 76 males and 76 females, and the overall median age was 35 years old (IQR: 28, 44). At baseline, 56 patients (36.8%) had macroscopic hematuria and 97 patients (63.8%) had hypertension. Based on the Oxford criteria, we classified the renal lesions as M1 (72% of patients), E1 (44.1% of patients), S1 (72.4% of patients), T1–T2 (23.7% of patients), or C1–C2 (35.5% of patients).

At the final follow-up, the overall TA-hematuria was 44.27 RBCs/μL (IQR: 13.43, 96.43) and the overall TA-serum albumin was 40.38 g/L (IQR: 36.5, 42.81). During the first 6 months of follow-up, 95 patients (62.5%) achieved proteinuria remission. (Table 2) 

### 3.2. Clinical and Pathological Characteristics of IgAN Patients with Persistent Hematuria 

We categorized the patients into two groups according to the degree of TA-hematuria (Table 1). The persistent hematuria group (>28 RBCs/μL) had 44 males and 50 females and a median age of 33 years old (IQR: 27.75, 44.25). The persistent hematuria group had a significantly higher percentage of patients with macroscopic hematuria (43.8% vs. 25.0%, *p* = 0.021), more severe microscopic hematuria (114 vs. 7.8 RBCs/µL, *p* = 0.001), and a lower level of hemoglobin (125.83 vs. 133.82 g/L, *p* = 0.017). However, the two groups had no significant differences in sex, age, BUN, serum albumin, uric acid, 24-h urine protein, TC, TG, serum C3, serum C4, or eGFR (all *p* > 0.05). Furthermore, separate analysis of the hemoglobin levels in males and females indicated no significant difference between those with persistent hematuria and those with hematuria remission in each analysis (Figure 2A).

Analysis of the pathological results indicated the persistent hematuria group had a significantly greater prevalence of segmental glomerulosclerosis, especially among females (Figure 2B), and a greater prevalence of crescents, especially among males (Figure 2C) (both *p* < 0.05). However, the two groups had no statistically significant differences in mesangial hypercellularity, endocapillary hypercellularity, and tubular atrophy/interstitial fibrosis (all *p* ˃ 0.05; Table 1). 

At the final follow-up, the persistent hematuria group had a significantly lower TA-serum albumin level (39.70 [IQR: 36.03, 41.84] vs. 41.68 [IQR: 37.41, 43.72] g/L, *p* = 0.022), compared with the hematuria remission group, which was still significant in females (Figure 2E). In addition, participants with persistent hematuria were more likely to develop ESRD and to attain the composite outcome (both *p* < 0.05). However, the two groups did not differ in receipt of treatments (RASS inhibitor, glucocorticoid, or immunosuppressant) or in achieving remission from proteinuria during the first 6 months (both *p* > 0.05). 

### 3.3. Long-Term Renal Outcome of IgAN Patients with Persistent Hematuria

During the mean follow-up of 58.08 ± 23.51 months, 15 patients (9.9%) reached the primary outcome of ESRD and 19 patients (12.5%) reached the combined renal endpoint. We performed Kaplan-Meier analysis to compare renal survival of patients with stratification by different patient characteristics. The results indicated lower renal survival rates in the group with macroscopic hematuria (*p* = 0.033, Figure 3A) and in the group with more severe TA-hematuria (*p* = 0.049, Figure 3B). We also stratified patients according to the composite of hematuria (persistence vs. remission) and proteinuria (persistence vs. remission). The results showed that the group with hematuria remission and proteinuria persistence had a better renal outcome than the group with hematuria persistence, but a poorer renal outcome than the group with hematuria remission and proteinuria remission (*p* = 0.001, Figure 3C). We then stratified patients according to the composite of TA-hematuria (high vs. low) and TA-serum albumin (high vs. low). The renal survival rate was the lowest in patients who had a combination of persistent hematuria and low TA-serum albumin (<40 g/L, *p* = 0.011, Figure 3D).

We performed Cox proportional hazards regression analysis to identify risk factors associated with the composite renal endpoint (D-SCr, or eGFR < 15 mL/min/1.73 m^2^). The univariate analysis showed that hypertension, macroscopic hematuria, more severe microscopic hematuria, BUN, hemoglobin, uric acid, eGFR, 24 h urinary protein, T1–T2, TA-serum albumin, and TA-hematuria were significantly associated with poorer prognosis (Figure 4A). The multivariable analysis (which controlled for all clinical characteristics significantly associated with renal survival) showed that more severe TA-hematuria (hazard ratio [HR] = 0.004, 95% CI: 0.001, 0.008; *p* = 0.010), TA-serum albumin level (HR = 0.845, 95% CI: 0.716, 0.998; *p* = 0.047, baseline eGFR (HR = 0.959, 95% CI: 0.927, 0.992; *p* = 0.017), and T1–T2 histology (HR = 12.038, 95% CI: 3.074, 47.136; *p* < 0.001) were independent risk factors for renal dysfunction (Figure 4B). 

We then performed ROC analysis to determine the diagnostic accuracy of TA-hematuria in predicting the progression of IgAN (Figure 5A). In all 152 patients, the area under the curve (AUC) for TA-hematuria was 0.697 (95% CI: 0.571, 0.823), the optimal cutoff value (based on Youden’s index) was 201.21 RBCs/μL, and the specificity was 92.1%, but the sensitivity was only 42.1%. A separate analysis of males indicated the AUC was 0.645 (95% CI: 0.462, 0.829), the optimal cutoff value was 37.25 RBCs/μL, the sensitivity was 77.8%, and the specificity was 56.7% (Figure 5B). In females, the AUC was 0.761 (95% CI: 0.595, 0.926), the optimal cutoff value was 201.21 RBCs/μL, the sensitivity was 60.0%, and the specificity was 87.9% (Figure 5C). These results thus demonstrated that the severity of TA-hematuria was a reliable prognostic indicator for the progression of IgAN.

## 4. Discussion

IgAN has highly variable clinical manifestations, and the progression of renal damage can occur slowly over a period of 10 to 20 years. More than 30% of Chinese adults with IgAN develop ESRD within 20 years [3], but some patients experience spontaneous resolution. Thus, early identification of risk factors that affect disease progression may allow for the implementation of appropriate interventions that can help slow the progression to ESRD and improve patient prognosis.

Patients with IgAN typically present with microscopic hematuria (although this is a non-specific sign) and occasionally with gross hematuria and upper respiratory or gastrointestinal infections [12]. We analyzed the clinicopathological characteristics and prognosis of IgAN patient according to the severity and persistence of hematuria. Among our 152 patients, 56 patients (36.8%) had macroscopic hematuria at baseline, and 94 patients (61.8%) had persistent hematuria at the end of follow-up. Our results indicated that the severity of hematuria after follow-up was strongly associated with the development of kidney failure (doubling of SCr or ESRD).

Previous studies confirmed that patients with IgAN who experienced episodes of gross hematuria had a more benign clinical course than those without gross hematuria [7,10,18,19,20]. In addition, a recent meta-analysis that examined 5158 subjects with IgAN concluded that macroscopic hematuria was associated with a decreased risk of ESRD [21]. Similarly, we found a higher cumulative renal survival rate in patients who had macroscopic hematuria at baseline. However, our multivariate Cox regression analysis showed that macroscopic hematuria was not a significant and independent prognostic factor. There is evidence that macroscopic hematuria is more common in patients presenting with early-stage IgAN, and less common in those with significantly impaired renal function (advanced-stage disease) [22]. Therefore, the value of macroscopic hematuria in predicting the progression of IgAN was limited, and it may only have predictive value in early IgAN patients. These results suggest it is more meaningful to monitor changes in microscopic hematuria during the progression of IgAN.

Recent studies indicated a lack of consistency regarding the prognostic value of microscopic hematuria in patients with IgAN. One study of IgAN patients found that mild microscopic hematuria increased the risk of ESRD during the subsequent 10 years [10]. Another study suggested that mild microscopic hematuria predicted renal deterioration in patients with IgAN who had eGFR values greater than 60 mL/min/1.73 m^2^ and no proteinuria [11]. However, other research suggested that patients with microscopic hematuria and minimal or no proteinuria had excellent outcomes, based on measurements of serum creatinine level [8]. There is other evidence that microscopic hematuria was unrelated to disease progression, based on measurements of proteinuria and ESRD [9]. Due to these conflicting results, a recent international risk prediction tool did not consider microscopic hematuria [23]. However, a study of 1 million young Israeli adults that had more than 20 years of follow-up found that persistent isolated microscopic hematuria significantly increased the risk of ESRD, and that 15% of subjects who developed ESRD initially had IgAN [24]. Our results indicated that gross hematuria and microscopic hematuria at the time of renal biopsy (baseline) were not independently associated with disease progression in patients with IgAN. However, we used TA-hematuria values to assess the severity of hematuria during follow-up, and our results showed that persistent hematuria was the strongest predictor of renal failure. Therefore, TA-hematuria appears to be a better prognostic indicator than hematuria measurements performed at a single time. 

Many recent studies have examined the use of TA-hematuria as a prognostic indicator for patients with IgAN. Some evidence indicated that TA-hematuria was a significant and independent risk factor for kidney failure [3]. Another recent European cohort study of 112 IgAN patients found that, during follow-up, TA-hematuria was strongly associated with kidney failure, and remission of hematuria was associated with a more favorable outcome [13]. A Chinese cohort study that examined 1333 patients with IgAN during 45 months of follow-up concluded that the severity of hematuria was independently associated with the progression of kidney disease, and persistent hematuria was associated with adverse kidney outcomes [14]. Another recent study showed that persistent hematuria might be an independent risk factor for poor renal outcomes (ESRD or 50% decline in eGFR) in patients with IgAN [21]. Our study of patients with IgAN confirmed that persistent hematuria was a statistically significant and independent risk factor for the progression of kidney disease. 

The results of earlier studies on whether hematuria is a risk factor for IgAN prognosis are debatable [7,8,9,10,11,12], likely as a result of the fact that these studies frequently use baseline values from a single urine test. Urine erythrocytes frequently change continuously with disease, and various studies use different detection methods, including the manual examination method (RBC/hpf) and the automated method (RBC/μL), but the detection of urine erythrocytes is frequently a point in time no matter which method was used. Based on the deficiency of the above studies, our study employed numerous point collections during follow-up to reflect the overall average value of the patient’s hematuria levels, which is more persuasive for the prognosis of IgAN. The technology currently used for automatic detection of urinary RBCs has matured and allowed us to routinely monitor TA-hematuria during follow-up. Our ROC analysis indicated that more severe TA-hematuria predicted poor renal outcome. This analysis indicated that at a cutoff value of 201.21 RBCs/μL led to a high specificity (92.1%) in predicting renal outcome. We discovered an intriguing difference between male and female patients with IgAN in the prognostic efficacy of TA-hematuria for prognosis, despite our efforts to avoid having women keep urine samples for testing during menstruation. This may be due to the structure of a woman’s urethra being different from a man’s, which may account for the significant difference in results. In additional, we found that the predictive value of TA-hematuria was higher in female patients (AUC = 0.761 vs. AUC = 0.645 in male patients), and the TA-hematuria values in female patients were closer to the predictive values in the overall population. As a result, we came to the conclusion that the TA-hematuria has a better clinical practice value in female patients with IgAN (optimal cutoff = 201.21 RBCs/μL).

Previous research proposed that the values of certain clinical parameters at the diagnosis of IgAN (hypertension and proteinuria [25]; decreased levels of eGFR, hemoglobin, and albumin [26]; and Oxford pathology scores (such as mesangial and endocapillary hypercellularity, segmental sclerosis, and tubular atrophy/interstitial fibrosis) were useful as predictors of disease progression [27,28]. Other studies found that IgAN with hematuria and minimal proteinuria was an indicator of progressive disease [29,30,31], and a recent study reported that remission of hematuria and proteinuria were markers of clinical remission in these patients [32]. One of our important results was that the effect of microscopic hematuria on kidney failure interacted with proteinuria. In particular, patients with microscopic hematuria who had persistent proteinuria (>1.0 g/day) had an increased risk of renal failure, consistent with previous research [14], and patients with microscopic hematuria with persistent hypoalbuminemia (TA-serum albumin < 40 g/L) also had a greater risk of renal failure. Therefore, we suggest that the presence of TA-hematuria, persistent proteinuria, and persistent hypoalbuminemia can be used to predict prognosis in patients with IgAN. However, additional studies are necessary to validate this approach.

The renal manifestations of glomerular hematuria-related pathology typically include tubules filled with erythrocyte casts and acute tubular necrosis, and these mainly occur during episodes of gross hematuria. The RBCs in the urine of IgAN patients with hematuria often release Hb and related products, and this is associated with oxidative damage, podocyte dysfunction, and fibrosis, and eventually leads to disruption of the glomerular filtration barrier [33,34,35]. Our data suggest that patients with persistent hematuria were more likely to have segmental glomerulosclerosis and to develop ESRD than those with hematuria remission. Therefore, the duration and severity of microscopic hematuria may be key indicators for monitoring renal function in patients with IgAN.

IgAN is an autoimmune disease in which disturbances in immune regulation are a central part of the pathogenesis of the disease. The mucosal immune system plays an important role in the progression of IgAN, and the frequent occurrence of hematuria is always complicated with upper respiratory or intestinal tract infections [36]. Our study found that, at the final follow-up, the persistent hematuria group had a significantly lower TA-serum albumin level. Patients with IgAN and persistent hematuria suggest active disease progression with increased autoantibody production and deposition of abnormally glycosylated IgA1 immune complexes in the glomerular tract, especially the complement bypass pathway activation, which can cause erythrocyte hemolysis and amplify the inflammatory cascade response, attracting more inflammatory cells and local chemokine production [37,38], leaving the patient in a persistent state of inflammatory infection. In the inflammatory state, the rate of albumin degradation is accelerated, and the rate of albumin synthesis is much lower than the rate of degradation, which in turn leads to a decrease in total systemic albumin [39]. Therefore, IgAN patients with persistent hematuria also exhibit persistent hypoproteinemia.

Our results also indicated that IgAN patients with persistent hematuria had significantly more crescent formation than those with hematuria remission. Similarly, a 2017 study found that the presence and the extent of crescents were independent predictors of outcome, and recommended modification of the Oxford MEST score to the MEDST-C score for classification of IgAN [15]. The correlation between the persistence of microscopic hematuria and the presence of crescents in IgAN suggests these two pathologies might have a causal relationship. An earlier study found that persistently high microscopic hematuria indicated continuing pathological activity that manifested as the formation of focal and segmental crescents [40]. This supports the use of persistent hematuria for predicting the progression of IgAN.

There are currently no specific therapies that can ameliorate the adverse effects of hematuria-associated adverse effects in patients with glomerular diseases. The only validated treatment for IgAN is the blockade of RAAS, specifically when proteinuria exceeds 1 g/day [41]. Fifty-seven of our patients (37.5%) used a RAAS inhibitor during the follow-up, but we have no evidence that this therapy decreased TA-hematuria. Therefore, prospective studies are needed to assess the effects of RAAS and other treatments that may impact persistent hematuria.

This study has several limitations. Firstly, we excluded patients who may have had IgAN, but did not receive renal biopsies. It is likely that these individuals had a lower incidence of persistent hematuria than the enrolled patients. Secondly, this was a single-center study that examined a relatively small number of IgAN patients. Therefore, a prospective multi-center study is required to further validate our results.

## 5. Conclusions

Our TA-hematuria measurements suggest that patients who have IgAN and persistent microscopic hematuria have a substantially higher risk of developing ESRD or impaired renal function, especially those who also have persistent proteinuria and persistent hypoalbuminemia. Our results also indicated that the severity of TA-hematuria could be considered an independent predictor for the progression of IgAN, especially in females. Therefore, when examining patients with IgAN, we suggest the inclusion of TA-hematuria measurements during follow-up in risk scoring systems that are used to predict renal outcome.

## Figures and Tables

**Figure 1 jcm-11-06785-f001:**
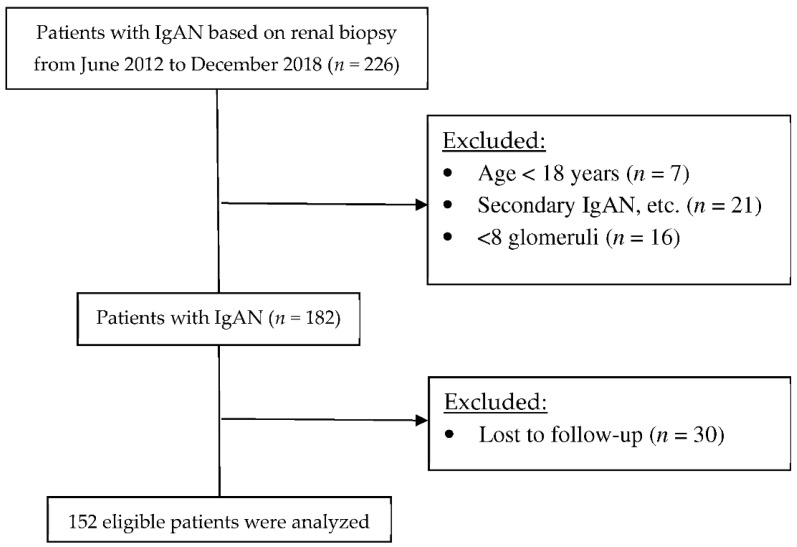
The study flow chart of the enrolment of IgAN patients.

**Figure 2 jcm-11-06785-f002:**
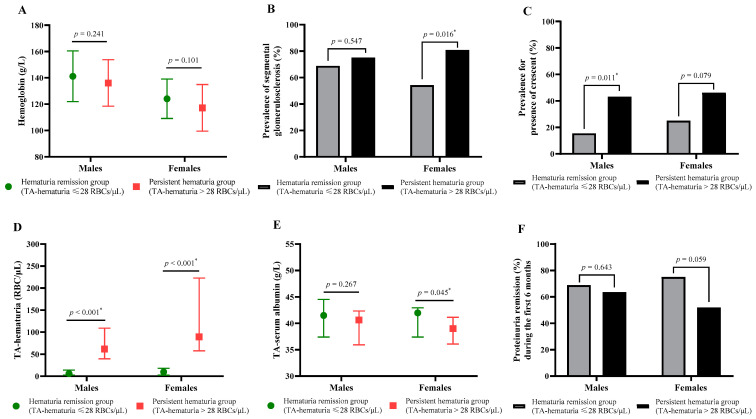
Comparison of clinicopathological data and the clinical data during follow-up and whether proteinuria remission of IgA nephropathy participants between hematuria remission and persistent hematuria groups in males and females. (**A**) Estimated glomerular filtration rate (eGFR); (**B**) the prevalence of segmental glomerulosclerosis (S1); (**C**) the prevalence for presence of crescent (C1–2); (**D**) TA-hematuria; (**E**) TA-serum albumin; (**F**) the present of proteinuria remission during first 6 months. *** Two-tailed *p* < 0.05.

**Figure 3 jcm-11-06785-f003:**
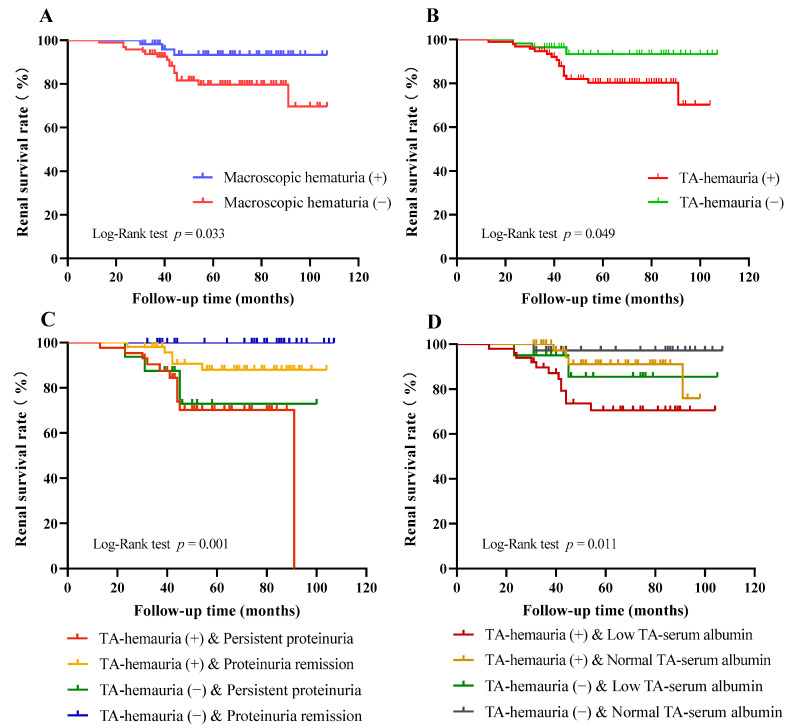
The renal survival rate of the Kaplan-Meier analysis. Kaplan-Meier curves comparison when dividing participants into strata. (**A**) History of macroscopic hematuria (macroscopic hematuria vs. no macroscopic hematuria) (*p* = 0.033). (**B**) Levels of time-averaged hematuria (higher TA-hematuria level vs. lower TA-hematuria level) (*p* = 0.049). (**C**) Composite TA-hematuria level and proteinuria remission (higher/lower TA-hematuria levels and positive/negative proteinuria remission) (*p* = 0.001). (**D**) Composite levels of TA-hematuria and TA-serum albumin (higher/lower TA-hematuria levels and higher/lower TA-serum albumin levels) (*p* = 0.011).

**Figure 4 jcm-11-06785-f004:**
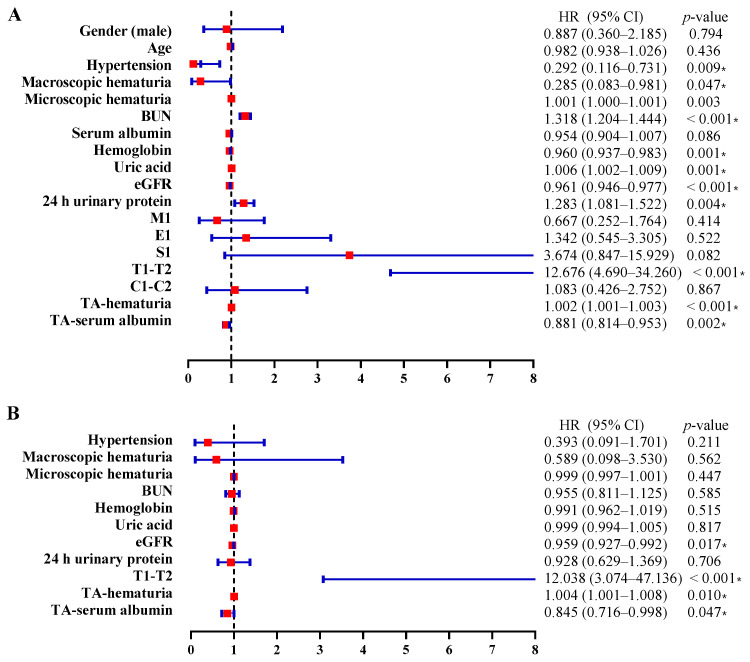
Risk factors for the renal outcome determined by (**A**) univariate or (**B**) multivariate Cox hazard analysis in IgAN. CI, confidence interval; HR, hazard ratio; BUN, blood urea nitrogen; TC, total cholesterol; TG, triglycerides; eGFR, estimated glomerular filtration rate; M1, mesangial hypercellularity; E1, endocapillary hypercellularity; S1, segmental glomerulosclerosis; T1–T2, tubular atrophy/interstitial fibrosis > 25%; C1–C2, presence of crescent; TA, time-averaged. * Two-tailed *p* < 0.05.

**Figure 5 jcm-11-06785-f005:**
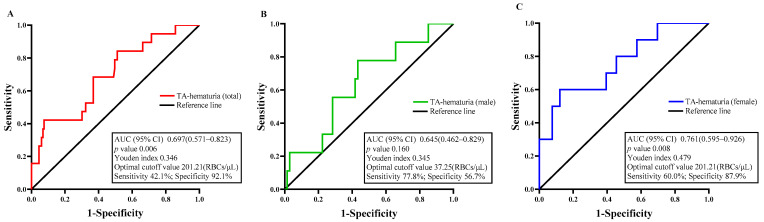
The value of the time-averaged hematuria (TA-hematuria) level in predicting renal prognosis. ROC curve analysis evaluates the predictive value of TA-hematuria level for the poor renal outcomes (**A**) in total participants, (**B**) in males, and (**C**) in females with IgA nephropathy.

**Table 1 jcm-11-06785-t001:** Baseline characteristics of participants with IgAN according to TA-hematuria.

	All (*n* = 152)	Persistent Hematuria during Follow-Up		*p*-Value
		Yes (Mean of >28 RBCs/μL) (*n* = 96)	No (Mean of ≤28 RBCs/μL) (*n* = 56)	
Clinical features				
Male sex (%)	76 (50.0)	44 (45.8)	32 (57.1)	0.179
Age (years)	35 (28, 44)	33 (27.75, 44.25)	35 (28.25, 44.75)	0.656
Hypertension (%)	97 (63.8)	63 (65.6)	34 (60.7)	0.543
Macroscopic hematuria (%)	56 (36.8)	42 (43.8)	14 (25.0)	0.021 *
Microscopic hematuria (RBCs/μL)	51.5 (13.0, 167.5)	114 (49.0, 274.0)	7.8 (2.08, 18.58)	0.001 *
BUN (mmol/L)	5.22 (4.11, 6.89)	5.05 (4.20, 6.83)	5.72 (3.93, 7.30)	0.476
Serum albumin (g/L)	38.55 (34.2, 41.65)	37.70 (33.93, 40.35)	40.15 (34.20, 42.70)	0.064
24 h urinary protein (g/d)	1.11 (0.44, 2.02)	1.17 (0.52, 2.10)	1.05 (0.39, 1.93)	0.578
Uric acid (μmol/L)	392 (314.3, 454.53)	388.10 (314.90, 456.55)	392 (314.00, 457.48)	0.795
Hemoglobin (g/L)	128.78 ± 20.08	125.83 ± 19.98	133.82 ± 19.38	0.017 *
TC (mmol/L)	1.32 (0.86, 2.01)	1.23 (0.84, 1.98)	1.52 (0.95, 2.24)	0.188
TG (mmol/L)	4.88 (4.15, 5.80)	4.81 (4.15, 5.82)	5.00 (4.16, 5.75)	0.410
Serum C3 (g/L)	0.99 ± 0.23	0.98 ± 0.23	1.00 ± 0.23	0.715
Serum C4 (g/L)	0.22 (0.18, 0.29)	0.22 (0.18, 0.29)	0.23 (0.19, 0.28)	0.644
eGFR (ml/min·1.73 m^2^)	90.5 (60.3, 115.5)	88.80 (59.88, 115.40)	91.65 (63.41, 117.33)	0.376
Oxford classification (MEST-C)				
M1	110 (72.4)	72 (75.0)	38 (67.9)	0.342
E1	67 (44.1)	43 (44.8)	24 (42.9)	0.817
S1	110 (72.4)	75 (78.1)	35 (62.5)	0.038 *
T1–T2	36 (23.7)	24 (25.0)	12 (21.4)	0.617
C1–C2	54 (35.5)	43 (44.8)	11 (19.6)	0.002 *

Data are expressed as numbers and percentages in non-continuous variables, as means ± standard deviation in parametric continuous variables, and as median and interquartile range in nonparametric continuous variables. BUN, blood urea nitrogen; TC, total cholesterol; TG, triglycerides; eGFR, estimated glomerular filtration rate; M1, mesangial hypercellularity; E1, endocapillary hypercellularity; S1, segmental glomerulosclerosis; T1–T2, tubular atrophy/interstitial fibrosis > 25%; C1–C2, presence of crescent; * Two-tailed *p* < 0.05.

**Table 2 jcm-11-06785-t002:** Follow-up data and outcomes of participants with IgAN according to TA-hematuria.

	All (*n* = 152)	Persistent Hematuria during Follow-Up		*p*-Value
		Yes (TA-Hematuria > 28 RBCs/μL) (*n* = 96)	No (TA-Hematuria ≤ 28 RBCs/μL) (*n* = 56)	
Follow-up				
TA-hematuria (RBCs/μL)	44.27 (13.43, 96.43)	75.30 (46.82, 158.51)	6.733 (2.56, 17.13)	<0.001 *
TA-serum albumin (g/L)	40.38 (36.5, 42.81)	39.70 (36.03, 41.84)	41.68 (37.41, 43.72)	0.022 *
Proteinuria remission during the first 6 months	95 (62.5)	55 (57.3)	40 (71.4)	0.082
Therapy status				
RAAS inhibitor (%)	57 (37.5)	40 (41.7)	17 (30.4)	0.165
Glucocorticoid (%)	97 (63.8)	61 (63.5)	36 (64.3)	0.927
Immunosuppressant (%)	121 (79.6)	78 (81.3)	43 (76.8)	0.510
Outcome				
Progression to D-SCr (%)	16 (10.5)	13 (13.5)	3 (5.6)	0.113
Progression to ESRD (%)	15 (9.9)	13 (13.5)	2 (3.7)	0.047 *
Composite outcome (%)	19 (12.5)	16 (16.7)	4 (7.4)	0.042 *

Data are expressed as numbers and percentages for non-continuous variables, as means ± standard deviation for parametric continuous variables, and as median and interquartile range for nonparametric continuous variables. TA, time-averaged; RAAS, renin-angiotensin-aldosterone system; D-SCr, doubling of serum creatine; ESRD, end-stage renal disease; The composite outcome was a composite kidney disease progression event defined as doubling of serum creatine or ESRD; * Two-tailed *p* < 0.05.

## Data Availability

The data that support the findings of this study are available from the corresponding author upon reasonable request.

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
