# Peer review of "Time-Averaged Hematuria as a Prognostic Indicator of Renal Outcome in Patients with IgA Nephropathy"

_jcm, 2022, doi:10.3390/jcm11226785_

Round 1

Reviewer 2 Report

Dear Authors

Hematuria in IgAN as a progressive disease marker is highily controversial in literature.

The authors showed that time-averaged ( TA ) persistent hematuria over 28 RBC  is associated with gross macrohematuria , low hemoglobin levels, low time-averaged  (TA) serum albumine, and more severe renal pathologic lesions.

They studied 152 IgAN Chinese patients along 58 months of follow-up.

Comments and Questions

Line  152.....  “ At the final follow-up the persistent hematuria group had significantly lower TA-serum albumine ...”    Why is that ?  Patients are not nephrotic so their urine albumine loss is minor.   Discuss any possible inflammatory mediator.

Line   182....fig  3 ...Kaplan-Meier analysis curves   ....A and B  are misplaced.

                                    Graphic colors ...red and brown (?)  are very similar.

Line   191,,,and fig  4A ... Considering hypertension and gross hematuria  the text and figure are not in agreement.    Please rephrase.

Line   208....  RBC  cutoff   value is different in males ( 37.2 ) and females ( 201.2 )    Please discuss the possible causes for that.

Line   282......” The methods used to measure RBC in urine differ among laboratories   ...” 

Please discuss about other methods and their impact in controversial results in literature .

Line   300......It is well known in literature that RBC release of hemoglobin and related oxidative damage cause podocyte dysfunction and eventually fibrosis.

In your paper females show more persistent TA hematuria than males (fig 2B). So, I could guess that IgAN  would be more aggressive in females than in males, which is not true in clinical practice.   Do you have  a explanation for that ?
